# Antimicrobial Peptides in the Battle against Orthopedic Implant-Related Infections: A Review

**DOI:** 10.3390/pharmaceutics13111918

**Published:** 2021-11-12

**Authors:** Bruna Costa, Guillermo Martínez-de-Tejada, Paula A. C. Gomes, M. Cristina L. Martins, Fabíola Costa

**Affiliations:** 1i3S–Instituto de Investigação e Inovação em Saúde, Universidade do Porto, Rua Alfredo Allen, 208, 4200-135 Porto, Portugal; bruna.costa@i3s.up.pt (B.C.); fabiolamoutinho@ineb.up.pt (F.C.); 2INEB–Instituto de Engenharia Biomédica, Universidade do Porto, Rua Alfredo Allen, 208, 4200-135 Porto, Portugal; 3FEUP–Faculdade de Engenharia, Universidade do Porto, Rua Dr. Roberto Frias, 4200-465 Porto, Portugal; 4Department of Microbiology and Parasitology, University of Navarra, Irunlarrea, 1, 31008 Pamplona, Spain; gmartinez@unav.es; 5Navarra Institute for Health Research (IdiSNA), 31008 Pamplona, Spain; 6CIQ-UP e Centro de Investigação em Química da Universidade do Porto, Departamento de Química e Bioquímica, Faculdade de Ciências, Universidade do Porto, 4169-007 Porto, Portugal; pgomes@fc.up.pt; 7ICBAS–Instituto de Ciências Biomédicas Abel Salazar, Universidade do Porto, Rua Jorge de Viterbo Ferreira 228, 4050-313 Porto, Portugal

**Keywords:** orthopedic implant-related infections, antimicrobial-peptides (AMPs), surface functionalization, peptide immobilization, peptide release, antimicrobial

## Abstract

Prevention of orthopedic implant-related infections is a major medical challenge, particularly due to the involvement of biofilm-encased and multidrug-resistant bacteria. Current therapies, based on antibiotic administration, have proven to be insufficient, and infection prevalence may rise due to the dissemination of antibiotic resistance. Antimicrobial peptides (AMPs) have attracted attention as promising substitutes of conventional antibiotics, owing to their broad-spectrum of activity, high efficacy at very low concentrations, and, importantly, low propensity for inducing resistance. The aim of this review is to offer an updated perspective of the development of AMPs-based preventive strategies for orthopedic and dental implant-related infections. In this regard, two major research strategies are herein addressed, namely (i) AMP-releasing systems from titanium-modified surfaces and from bone cements or beads; and (ii) AMP immobilization strategies used to graft AMPs onto titanium or other model surfaces with potential translation as coatings. In overview, releasing strategies have evolved to guarantee higher loadings, prolonged and targeted delivery periods upon infection. In addition, avant-garde self-assembling strategies or polymer brushes allowed higher immobilized peptide surface densities, overcoming bioavailability issues. Future research efforts should focus on the regulatory demands for pre-clinical and clinical validation towards clinical translation.

## 1. Introduction

As life expectancy grows worldwide, the need to repair or replace fragile/fractured bone and joints, through orthopedic surgery, increases [1]. Annually, approximately 1.5 M joint arthroplasties are performed in Europe, and 1 M in the United States [2,3]. Infection is still one of the major problems associated with implant failure [4,5,6]. The low infection rates of 1–2.5% reported for primary knee and hip replacements translate into a very high humane (substantial morbidity and 1-year mortality rate of 8–25.9%), and economic, burden (exceeds the cost of 354k € per case) [4,5,6]. Moreover, infection rates are more frequent in case of trauma (around 30%) or upon second revision surgeries (up to 25%) [7]. Similarly, dental implants may be affected by bacterial colonization, causing peri-implant mucositis and peri-implantitis, resulting in bone mass loss, and hindered osteointegration, affecting up to 50% of implant sites [8,9].

Different factors favor bacterial colonization of implants including: (i) the surgery itself, allowing a direct entry way and inducing inflammatory escalation, (ii) the implant itself, that causes the metabolic exhaustion of neutrophils, which become less efficient in removing bacteria [10] and whose conditioning layer serves as anchoring points for bacteria, and (iii) the low blood vessel density in the vicinity of the implant, which prevents a timely arrival of immune cells and antibiotics [11,12].

On the other hand, bacterial colonization can by itself hinder implant tissue integration [12], particularly when biofilms are formed on implant surfaces. Indeed, after their initial adhesion, bacteria can form microcolonies that produce extracellular molecules creating a protective matrix, which can further evolve and grow towards a mature biofilm [13,14]. Biofilm-forming organisms are, therefore, protected from the host immune system and from antibiotics [13]. The microenvironment inside biofilms leads to the generation of an isogenic subpopulation of antibiotic-tolerant bacteria, called persister cells (for more information, see reference [13]). Furthermore, switching to the biofilm growth mode facilitates horizontal gene transfer between bacteria (even resistance genes), therefore promoting antibiotic resistance [13,14,15] (further information on bacterial evasion mechanisms in reference [16]).

The biofilm induced antibiotic resistance explains why the treatment of prosthetic joint implant-associated infections (PJIs) is so difficult, as the chance to eliminate biofilm from the implant is below 50%, even after prolonged antibiotherapy [17].

More complex PJIs preventative measures have been progressively applied in clinical practice, including perioperative and postoperative strategies [18]. However, in some cases, infection develops and current treatment includes antibiotherapy, which relies on systemic administration of a combination of antibiotics, such as rifampicin, daptomycin, fluoroquinolones, vancomycin, or amikacin [19]. However, due to the poor peri-implant vascularization, local antibiotic concentration is frequently insufficient to control the infection [20]. Therefore, in most cases, the removal of the implant is key, imposing an invasive procedure with tissue debridement, contributing to further morbidity and additional revision surgeries [21].

Alternatives to maximize target tissue concentration and minimize systemic toxicity risk include local administration of antibiotics in the form of antibiotic-loaded cements, carriers or coatings, to complement the systemic antibiotic treatment [22,23,24]. Despite the fact that use of antibiotic-loaded cement has notably increased, negative effects on the mechanical stability of the cement have been reported [25]. In addition, the possible local and systemic toxicity of the loaded antibiotic, and the development of antibiotic resistant bacterial strains, were not completely solved by these strategies [25,26]. Unfortunately, an astonishing 50% of PJIs are caused by Methicillin-Resistant *Staphylococcus aureus* (MRSA) [16], and up to 40% of *Staphylococcus epidermidis* and 32% of *S. aureus* strains isolated from orthopedic-related infections are resistant to gentamicin [10]. Knowing the scarcity of new antibiotics in the pipeline over the last three decades, alternatives for conventional antibiotics are urgently being pursued [27,28,29,30].

Antimicrobial peptides (AMPs) constitute a promising class of molecules that could potentially overcome antibiotic resistance [28,29,31,32]. AMPs are part of the innate immune system of many organisms, having a broad-spectrum antimicrobial activity (including activity against ESKAPE (*Enterococcus faecium*, *S. aureus*, *Klebsiella pneumoniae*, *Acinetobacter baumannii*, *Pseudomonas aeruginosa*, and *Enterobacter* sp.) pathogens), high efficiency at low concentrations, high anti-biofilm activity, and even immunomodulatory potential [33,34,35,36]. To date, more than 3000 AMPs have been characterized (Peptide Database: http://aps.unmc.edu/AP/main.php, accessed on 10 September 2021) and only seven of them have been approved by the U.S. Food and Drug Administration (FDA), mostly for topical applications or to systemically treat severe bacterial infections [37,38].

Due to their mode of action, which involves targeting the bacterial membrane or affecting multiple targets within bacteria, AMPs are much less likely to induce resistance than conventional antibiotics (see more details on AMPs mechanisms of action in references [39,40]). AMPs can be incorporated into coatings to be applied onto implant surfaces or can be immobilized onto biomaterial surfaces, thereby providing antimicrobial action directly at the implant site, where it is most needed. These strategies circumvent some of the limitations inherent to freely circulating AMPs, which display a short half-life (due to enzymatic proteolysis or sequestration by blood proteins), as well as cytotoxicity at high concentrations [41,42].

The present review focuses on the potential of AMP-based solutions to address orthopedic implant-related infections (Figure 1). The review is divided in two main parts: the AMP-releasing systems, comprising bone cements or beads, and the releasing systems from titanium-modified surfaces; and AMP immobilization strategies, covering different methods of AMP coupling to titanium or to other model surfaces with potential of translation for surface coatings.

In this regard, we discuss the methodologies used, the antibacterial activity achieved, and the issues that each strategy presents, as well as the further steps that are needed to complete preclinical validation.

The goal of this review is to offer an historic overview from the past to the state of the art developments in AMP-based strategies designed for the orthopedic field.

## 2. Antimicrobial Peptides (AMPs)-Releasing Systems

Antimicrobial-releasing surfaces are designed to hinder bacterial adhesion and proliferation both on the implant itself and on the surrounding tissue, which is an important niche for bacterial survival [43]. An efficient release system should provide a fast initial release of the drug (AMPs) within 6 h post-implantation at effective concentrations to avoid bacterial proliferation [44], followed by prolonged local release at sufficiently high concentrations to eradicate any residual bacteria [44,45]. An inherent disadvantage of releasing systems is the limited capacity of the drug (AMPs) stock, which compromises its long-term efficacy [46]. Moreover, the released AMPs are susceptible to inactivation due to aggregates formation or protease-mediated digestion, decreasing their bioavailability [47]. Furthermore, increasing AMP concentration to counteract a reduced bioavailability is hampered by AMPs toxicity at high concentrations, so an equilibrium must be found that ensures AMPs antimicrobial activity while preserving tissue biocompatibility. Nevertheless, local delivery is a valid solution to afford high local efficiency against infection. Several AMP-releasing systems dedicated to orthopedic implants have been described in the last few decades [27,29,32,48,49,50,51,52,53,54,55,56,57,58,59,60,61,62,63,64].

### 2.1. Bone Cements/Beads

Bone cements (calcium phosphate and poly(methyl methacrylate)) have been used for years, to stabilize bone and implants, and also as antibiotics carriers [65,66,67]. Antibiotics loaded into bone cements could be easily replaced by AMPs, representing a possible straightforward strategy to step into the market [48,49,50,51].

Calcium phosphate, Ca_3_(PO_4_)_2_ (hereafter referred to as CaP), has drawn attention for orthopedic applications, since it can both enhance bone growth onto the implant [68] and be absorbed by the bone tissue, thereby assisting its repair. Therefore, the combination of CaP and an appropriate AMP could promote osteoconductivity while also affording antimicrobial activity. Stallmann et al. studied the combination of human lactoferrin 1-11 (hLF1-11) (complete AMP sequences can be found in the Appendix A) with six commercially available CaP bone cements (Biobon (Biomet Merck Biomaterials, Darmstadt, Germany), Calcibon (Biomet Merck Biomaterials), Biofil (experimental, DePuy CMW, Blackpool, UK), Bonesource (Stryker-Leibinger, Freiburg, Germany), Chronos Inject (experimental, Mathys, Bettlach, Switzerland), and Norian SRS (Mathys). The peptide release profiles obtained were similar between ceramic cements, having a two-phase release (i.e., initial burst-release and gradual sustained diffusion). However, the amount of released AMP depended on the specificities of the cement, namely the granulometry, and related porosity of the powders. Biobon, Biofil, and Chronos cements released significantly more hLF1-11 than the other three cements, presenting a higher burst peptide release in the first 24 h, up to 33.8% of the loaded AMP.

Interestingly, the antimicrobial activity was not directly related to the amount of peptide released from the cements. All six cements killed 83–98% of MRSA after 24 h, meaning that even the cement with the lowest hLF1-11 content (Bonesource; 0.095 mg/g) provided sufficient bactericidal activity [54]. These promising results prompted the same research group to evaluate the Bonesource CaP cement loaded with hLF1-11 (50 mg/g) in a *S. aureus* rabbit osteomyelitis prevention model (femur osteomyelitis) [53]. This was the first study describing the in vivo efficacy of a locally released AMP in the prevention of osteomyelitis. Stallmann et al. then performed a second assay to study the in vivo release and immunomodulatory effects of peptide hLF1-11, and the osteointegration of the cement in the femoral canal of a rabbit model. They reported an almost complete peptide release within 7 days, with tissue ingrowth into the cement and no signs of inflammation or necrosis, justifying the use of hLF1-11 as a prophylactic agent [52].

Faber et al. went a step further through the validation of hLF1-11 in a chronic osteomyelitis rabbit model (at tibia) against an MRSA strain isolated from a patient with osteomyelitis [51]. For that, a different bone cement (Calcibon) was loaded with either 50 mg/g of hLF1-11 or gentamicin. The hLF1-11 treatment was comparable to that of gentamicin, not only significantly reducing bacterial load compared to controls but also significantly diminishing the radiological and histopathological scores [51].

Alternatively, CaP was also explored as a coating by electrolytic deposition onto titanium (Ti) surfaces, creating microscopical porosity and large surface area for loading an AMP, namely Tet213 (an HHC36 derivative with an extra Cys at the *C*-terminus) [57]. This CaP coating, loaded with up to 9 µg of AMP/cm^2^, showed no cytotoxicity against MG-63 osteoblast-like cells, revealing antimicrobial activity against both Gram-positive (*S. aureus*) and Gram-negative (*Pseudomonas aeruginosa*) bacteria with 6 log reductions within 30 min, as assessed by measuring colony-forming units (CFU) [57]. The same authors demonstrated that CaP-Tet213 was more efficient than the coatings made with the commercially available AMPs MX-226, hLF1-11, or the antibiotic tobramycin (CaP-MX226, CaP-hLF1-11, or CaP-tobramycin) at equimolar concentrations of Tet213 [57]. CaP-Tet213 retained its potency after repeated uses while remaining biocompatible to osteoblast-like cells [57]. Nevertheless, the osteoconductivity and antibacterial performance of CaP-Tet213 remains to be verified in an animal model.

Later, Kazemzadeh-Narbat et al. compared two AMPs (HHC36 and Tet213) in CaP coated-Ti implants, in terms of their in vitro biocompatibility. Tet213 exhibited higher cytotoxicity at lower concentrations compared to HHC36. Therefore, CaP-HHC36 was further studied in terms of release rate and bactericidal profile. The in vitro evaluation revealed a high release rate at the early time points, followed by a slow and steady release for days (coating released 71% of AMP in the first 30 min and 91% within 24 h), reporting a total of 34.8 ± 4.2 µg of AMP/cm^2^ loaded into the coating [59]. Compared to the previous study, this appears to be a major improvement in release profile; however, different methodologies were applied in this measurement (as the loading process appears to be the same), which may artificially produce this significant difference. Importantly, CaP-HHC36 killed 100% of *S. aureus* and *P. aeruginosa* in less than 150 min, without negatively affecting MG-63 osteoblast-like cells [59]. Regarding the in vivo evaluation, a significant bone on-growth on CaP-HHC36 was verified compared to the Ti surfaces (~60% versus ~36%). However, since the animal study did not involve an infection model, the in vivo efficacy of the AMP-loaded implants needs to be confirmed.

Poly(methyl methacrylate) (PMMA) was considered the gold standard in bone cements [65]; however, its subsequent removal is required in some applications, which explains why biodegradable cements as CaP have been gaining relevance [65]. Nevertheless, PMMA bone cements were also targeted as potential AMPs carriers. Taking into consideration that PMMA polymerization process is exothermic [65], it is important to ensure that this does not negatively affect AMPs activity or its long-term release profile [29,49,50,55].

In 2003, Faber et al. tested the release of peptide Dhvar5 from PMMA beads in vitro. Different Dhvar5 amounts were loaded (120, 600, or 1200 μg), which correlated positively with the AMP release kinetics [50], achieving 91% release at 1200 μg load. This result may be explained by increased porosity of the carrier matrix at higher peptide concentrations [50], culminating in release profiles higher than those previously reported for antibiotics [69,70]. Released Dhvar5 remained biologically active against a clinical MRSA isolate. This system was later validated in vivo in comparison with gentamicin-containing beads, confirming the antimicrobial activity of Dhvar5 against MRSA, although at a lower efficacy level than gentamicin [49].

More recently, Melichercik et al. demonstrated that dodecapeptides of the halictine-2 series loaded into PMMA cement, efficiently prevented microbial adhesion and subsequent biofilm formation on the cement surface [55]. Selected derivatives (H27 and H39) were then tested ex vivo in the spongy part of infected or non-infected human bone samples, using bone cement with or without AMPs, and were compared to cement with antibiotics (vancomycin or gentamicin). The two AMPs proved their superiority to current antibiotics against a set of both American Type Culture Collection (ATCC) and osteomyelitis clinical isolates (including MRSA) when examined in this induced osteomyelitis ex vivo model [55]. However, one limitation of this study is that the extent of penetration of AMPs and antibiotics into the surrounding bone tissue, as well as their release kinetics from the bone cement to the tissue, were not directly assessed.

Later, four of these halictine-2 series analogues were used by Volejníková et al. for comparison with vancomycin in PMMA beads [29]. The peptides were designated as H27, H27D, H39, and H39D, where H27D and H39D are, respectively, the D-analogues of H27 and H39, included to prevent possible degradation of AMPs by bacterial proteases. These D-analogues did not show significant differences in inhibiting bacterial adhesion when compared to their parent peptides (H27, H39). Adding to the four analogues, H39LD, with a syndiotactic sequence of the H39 analogue, was created. The release kinetics of both AMPs and vancomycin occurred with an initial burst release (first hours), followed by rapidly slowing release rates throughout the next 3 days, with negligible incline during the following 10 days [29]. The H39 cement showed a noticeably higher release rate (~27% over 14 days) than the cement loaded with vancomycin (17%) [29]. Both AMPs and vancomycin beads were tested with bacterial strains from both ATCC and osteomyelitis clinical isolates (including MRSA) [29]. Overall, the PMMA model implants (beads) loaded with either of the five AMPs achieved a 5 log mean reduction in bacteria adhered to the surfaces of implants, having broader antibiofilm activity than vancomycin. However, neither the AMPs nor vancomycin were able to eliminate the planktonic bacteria in the media surrounding these beads. Nevertheless, the AMPs developed by Melichercik et al. and Voleijnikiva et al., in particular, peptide analogues H27 and H39, demonstrated to be good candidates by showing broad-spectrum activity and ability to withstand PMMA polymerization.

Overall, the releasing rates and antimicrobial activities of AMP-loaded PMMA or CaP beads/cements are similar or even better when compared with their antibiotics-based counterparts, which encourages the establishment of this strategy [71,72]. Therefore, future in vivo studies, with adequate animal models and more realistic scenarios, will allow to assess their potential application in orthopedic surgery. Different osteomyelitis models to assess in vivo performance of titanium implants have been recently reviewed [73,74].

### 2.2. Titanium-Based Releasing Systems

Metals used in orthopedic implants include surgical grade stainless steel (commonly 316L), cobalt-chromium (Co-Cr) alloys, and pure commercial titanium (Ti) or titanium alloys (e.g. Ti_6_Al_4_V). Pure titanium and its alloys are the most commonly used materials for permanent implants in contact with bone due to high biocompatibility and good corrosion resistance [75].

Titanium-based releasing systems have been used to minimize bacterial adhesion, inhibit biofilm formation and provide effective bacterial killing to protect implanted biomaterials [76]. In particular, AMP-releasing coatings on orthopedic and dentistry implants have been explored in references [27,32,56,57,58,59,60,61,62,63,64,77].

#### 2.2.1. TiO_2_ Nanotubes Arrays

Titanium dioxide (TiO_2_) is a frequent modification of Ti orthopedic materials, as it offers a more osteoconductive interface [78]. TiO_2_ can also be used in the fabrication of nanotubular structures by the anodization method, which have attracted wide interest as drug carriers due to their high surface-to-volume ratio and controllable dimensions [78].

Ma et al. were the first to evaluate the applicability of self-organized and vertically oriented TiO_2_ nanotubes as delivery systems for the AMP HHC36, at 2 mg/mL [60]. HHC36 presented a high release rate (5 µg/h) in the first 4 h, followed by a steady and relatively slow release in the following 7 days [60]. This system exhibited in vitro bactericidal activity against *S. aureus* in the liquid surrounding the nanotubular surface and reduced in 2.3 log the surface bacterial colonization, despite not avoiding it [60]. Similarly, Li et al. [27] developed, in 2017, TiO_2_ nanotubes containing the AMP GL13K (GL13K-TNTs), aimed at preventing infection on dental implants. The amount of the loaded AMP onto the nanotubes (10 mg/mL) was 5-fold higher than of the previously reported material [60]. This system presented a high release rate in the initial 30 min and successfully prevented the growth of common dental etiologic agents: *Fusobacterium nucleatum* and *Porphyromonas gingivalis,* as assessed by an in vitro disk-diffusion assay [27]. Additionally, GL13K did not inhibit mammalian cell proliferation nor enhanced macrophage inflammatory responses [27].

Equally applying TiO_2_ nanotubes and HHC36 (~1 mg/mL), Kazemzadeh-Narbat et al. developed a multi-layered coating composed of vertically oriented TiO_2_ nanotubes, a thin layer of CaP, and a palmitoyl oleoyl phosphatidyl choline phospholipid (POPC) film [58]. According to the authors, this system should allow a controlled release of HHC36 from the surface, avoiding a strong burst release that would culminate in depletion of the loaded AMP. In addition, the system would offer a dual beneficial effect, i.e., antimicrobial and osteoconductive, attributed to the thin layers of TiO_2_ nanotubes and CaP coatings impregnated with AMPs and the bioinspired cell membrane, such as POPC film. They reported a slow and steady release of the AMP for at least 72 h (cumulative release of 150 µg), which proved to be highly effective against both Gram-positive (*S. aureus*) and Gram-negative (*P. aeruginosa*) bacteria [58]. This system had its biocompatibility evaluated, showing no toxicity to osteoblast-like cells (MG-63), while promoting mild platelet activation and adhesion on the implant surface, and causing very low hemolysis.

More recently, Chen et al. [61] developed eight HHC36 (1.5 mg/mL)-based TiO_2_ nanotube systems with a switchable response dependent on bacterial presence (smart releasing coating) (Figure 2). To this end, the surface of the TiO_2_ nanotubes was firstly modified with dopamine, whose amine groups were covalently linked to carboxyl groups of poly(methacrylic acid) (PMAA) (Figure 2A). PMAA is a pH-sensitive polymer that collapses at pH < 6, allowing the opening of the “gate” of nanotubes. Therefore, AMP loading was performed starting at low pH and then rising it up to 7.4 (physiological environment), to promote the swelling of PMAA and consequent encapsulation of the AMPs, by “gate closure”. In other words, at this pH, the nanotubes remain closed, acting as an AMP reservoir capable of extending the release time from dozens of hours up to 10 days [61]. When bacterial infection occurs in the implant, pH in its vicinity drops below 6, leading to collapse of PMAA, i.e., to the opening of the “gate”, hence triggering rapid release of the AMPs from the nanotubes for immediate bacterial killing (Figure 2B).

This system showed high bactericidal activity (killing > 2 logs of bacteria in comparison with the control) against *S. aureus* (including an MRSA strain), *Escherichia coli*, and *P. aeruginosa*. In addition, improved biocompatibility regarding human bone mesenchymal stem cells (hBMSCs) and osteogenic activity were reported in vitro [61]. These results were validated in vivo in the bone defect New Zealand rabbit model, and authors claimed that bacterial infection during the acute infection period after implantation was fully prevented, without compromising biocompatibility [61].

More recently, mesoporous titania-covered titanium implants were explored to design a delivery system for the Proline And Arginine Rich End Leucine Rich Repeat Protein (PRELP)-derived AMP, RRP9W4N [79]. This system had a maximum loading capacity of 650 ng/cm^2^ after 2.5 h and allowed a sustained release of the AMP (only 18% of AMP was released after 20 h) [79]. The system presented equal or even better antibiofilm properties (93% reduction of biofilm coverage and 89% reduction of biofilm amount against *S. epidermidis* Mia strain) than the clinically used antibiotic cloxacillin (90% and 42% reductions, respectively). In addition, no negative effects on osteointegration in vivo were observed [79]. Still, despite mesoporous titania having been suggested as a good candidate for AMP releasing systems, in vivo antimicrobial tests are needed in the future.

#### 2.2.2. Polymer Coatings for Ti-Based Surfaces

Polymers, particularly hydrogels, are another promising type of material for the development of AMP-releasing coatings to use on Ti-based systems [20,80]. Besides working as carriers, some such polymers also display antimicrobial effects on their own, potentially providing additive or even synergistic antimicrobial effects when used in combination with AMPs [20].

Mateescu et al. [62] have developed two different alginate-catechol-based hydrogels embedding the AMP Cateslytin (CTL) at a final concentration of 200 μM. These hydrogels were adhered to Ti surfaces, thanks to catechol strong adhesion properties [81]. AMP release was reported to happen at least over 48 h. Authors suggested that, as the gel remained stable over at least 28 days, it is probable that some CTL might be slowly released during several weeks. The observed release was associated with 100% inhibition of *P. gingivalis* growth after 24 h, without any signs of toxicity against human gingival fibroblasts [62]. Considering that CTL is active against a large variety of pathogens (including non-hemolytic *S. aureus*), further studies could clarify the potential of such coating against a broader spectrum of bacteria and wider range of applications [82].

Cheng et al. proposed a gelatin-based hydrogel loaded with the AMP HHC36 (1.0 mg/mL) to adhere to Ti surfaces [56]. This hydrogel allowed a burst release of AMP to reach a cumulative release of 37% within the first 24 h, followed by a relatively steady release of AMP over the next 20 days [56]. The authors reported a significant reduction (several logs) in counted CFUs at 4 h and complete eradication of bacteria (*S. aureus*, *S. epidermidis*, *E. coli*, and *P. aeruginosa*) at 24 h [56].

Bormann et al. coated Ti-surfaces with Poly(d,l-lactide) and incorporated the antimicrobial peptide AMP2 (10% *w*/*w*, 20% *w*/*w*, or 30% *w*/*w*). When challenged with *S. aureus* in a zone inhibition test, these surfaces were capable of inhibiting bacteria growth in a dose dependent manner, however, to a less extent than gentamicin [83]. Further studies about the release profile of this AMP and in vitro assays using the Ti surfaces are needed to confirm the potential of this material.

More recently, Rodriguez-Lopez et al. [32] coated medical-grade Ti disks with crosslinked layer-by-layer (LbL) chitosan/hyaluronic acid (CH/HA) hydrogels releasing β-peptide (0.44 mg/mL) gradually at a constant rate of 4.6 ± 2.2 µg/cm^2^/day over a period of 28 days. These hydrogels could reduce in 60% the biofilm formed by *S. aureus* for up to 24 days and resisted five separate bacterial challenges over 18 days. When hydrogel was tested with preosteoblast cells (MC3T3-E1), no toxicity was observed, which demonstrates the selectivity of the β-peptide loaded films. This strategy afforded a longer release period as compared to previous strategies, potentially offering protection from bacterial attachment and biofilm formation over extended time periods.

The LbL technique was also employed by Shi et al. [64]. The authors coated a pure Ti surface with CH and then a LbL of HA acid and Tet213-collagen IV hydrogels. The broad-spectrum AMP Tet213 was linked to collagen IV via sulfosuccinimidyl 4-(*p*-maleimidophenyl)butyrate (sulfo-SMPB) (Figure 3).

The degradation of this multilayer system inhibited the growth of *S. aureus* and *P. gingivalis* after 24 h, as well as prevented early *S. aureus* biofilm formation in vitro [64]. Moreover, its long-term sustained AMP release (28 days) was able to maintain antimicrobial activity throughout the healing period after implant placement [64]. Biocompatibility assays showed no early cytotoxicity against HaCaT cells nor erythrocyte hemolysis [64]. The authors aim to perform future in vivo comparison studies against products available in the market to treat dental implant-related infections (peridex, tetracycline fibers, or minocycline hydrochloride ointment).

A different technology developed for the sustained release of antimicrobials is Polymer-Lipid Encapsulation Matrix (PLEX) coatings. PLEX coatings are multi-alternating layers of polymer and biodegradable lipid, containing polylactic-*co*-glycolic acid, dipalmitoyl phosphatidyl choline, and distearoyl phosphatidyl choline, capable of a sustained release of entrapped drugs during 3–4 weeks [84,85,86]. De Breij et al. firstly proved the efficacy of PLEX strategy in the release of an AMP (OP-145, a LL-37-derived AMP) that was reported to cause a bacterial load reduction of 38% on an humerus intramedullary nail infection rabbit model [77]. More recently, Riool et al. combined two OP-145 derivatives (SAAP-145, and SAAP-276) [63] with PLEX strategy, which allowed reduction of viable MRSA in the implant and peri-implant soft tissue in mice. This is a promising result that had not been observed during AMPs injection, along with subcutaneous implants in mice, nor with doxycycline-PLEX coated implants [43]. Both studies reported a controlled and prolonged release profile (55% of AMP released during the first 48 h) followed by a sustained release for up to ~20 days.

As inferred from the studies above discussed, a promising milestone has been accomplished, offering in vivo evidence of anti-infection activity of released AMP (either from beads or coatings) during extended periods (>20 days). Multi-layer and smart-releasing coatings that deliver AMPs upon a trigger, appear to be the best option to overcome limited drug reservoir capacity and bioavailability issues associated with these systems. Noteworthy, many authors showed preference for use of the AMP HHC36 (KRWWKWWRR). This particular AMP was applied in CaP coatings, TiO_2_ nanotubes, multilayer coatings, and hydrogels [56,58,60,61,64]. The reason for this widespread use of HHC36 might be its unique features: broad-spectrum activity, including against multidrug-resistant “superbugs”, at very low concentrations (0.3 to 11 µM), and very low cytotoxicity, including minimal hemolysis at concentrations up to 251 µM, allied to a very short amino acid sequence (9 residues) [56,87,88]. HHC36, therefore, offers optimal bioactivity along with a reduced and more transferable production cost [56,87,88].

## 3. AMPs-Grafted Systems

Implant coatings based on immobilized AMPs have many advantages, such as long-term stability and low toxicity, in comparison with other approaches that rely on leach- or release-based systems [56,89,90,91,92]. However, since the antimicrobial activity is restricted to the surface of the implant, this strategy does not address bacteria in the surrounding tissue, which can be source of infection. In addition, proteins, blood platelets, and dead bacteria can also block the antimicrobial groups on the surface [43,93]. Thus, the antimicrobial activity of the resulting coatings may be strongly reduced compared to the activity of the peptide in free form [43,93].

There are several parameters to be considered in performing a successful immobilization of active AMPs, such as the maintenance of the AMP antimicrobial structural features after immobilization [43], which is influenced by the spacer applied (length, flexibility), as well as by the orientation and surface density of the immobilized peptide [92]. Depending of chemical tethering procedure and the orientation of the grafted AMP, the antimicrobial activity of the surface with immobilized AMPs can change [92].

The coupling strategies to obtain, for example, Ti-AMP implants, range from different surface functionalization (e.g., silanization) and subsequent peptide grafting chemistries, to the application of chimeric peptides, or even to the making of AMP-grafted polymeric coatings.

### 3.1. AMPs Tethering onto Silanized Surfaces

Silanization is a surface modification strategy, which takes advantage of existing or introduced surface hydroxyl groups that can be stably linked to silicon atoms [94]. It is commonly used in Ti, hydroxyapatite, and many other metal oxide surfaces [94].

Several studies report on AMP immobilization onto silanized Ti, having important differences in terms of hydroxyl introduction method, reaction time, temperature, solvent, post-silanization thermal curing, and concentration of the silanization agent (Table 1) [94,95,96,97,98,99]. This is not surprising, since silanization reaction parameters strongly influence the stability of the formed linker layer, which can result on a thick polymerized silane network subjected to hydrolysis in certain conditions [100]. The chosen silanization agent introduces the proper reactive groups (e.g., amine or carboxyl groups) for surface conjugation with AMPs.

Gabriel et al. were the first authors to describe silanization as a method of covalent binding of AMPs onto Ti. In this work, a Ti surface pre-treated with piranha solution was silanized with 3-aminopropyltriethoxysilane (APTES) or glycidyloxypropyl triethoxysilane (epoxy silane) to compare chemoselective (by either its *N*- or its *C*-terminus) to random coupling of the AMP (human cathelicidin LL-37, 0.5 mg/mL). These authors also tested the introduction of a polyethylene glycol (PEG) spacer instead of direct coupling [98] (Figure 4).

They reported that chemoselective immobilization in combination with a PEG spacer was of paramount importance for bactericidal activity against *E. coli*, as randomly immobilized LL-37, both with and without the use of a spacer, had no activity. The authors postulated that PEGylated Ti surfaces grant lateral mobility to surface-bound AMP, while *N*-terminal conjugation allowed parallel orientation of the peptide helices, likely facilitating the interactions with bacterial membranes [98]. More recently, shorter derivatives of LL-37, namely KR-12 (LL-37 fragment 18-29) and FK-16 (LL-37 fragment 17-32), were also immobilized onto Ti [89,99]. In both cases, alkali-treated Ti was silanized with APTES, followed by direct KR-12 (1 mg/mL) immobilization [89], or by spacer-supported FK-16 (2.5 mg/mL) immobilization through the short bifunctional crosslinker 6-maleimidohexanoic acid [99]. Nie et al. [89] reported high CFU reduction (90% for *S. epidermidis* and methicillin-resistant *S. epidermidis* (MRSE)), without exhibiting cytotoxicity to hBMSCs (allowing even proliferation). Mishra and Wang [99] reported broad-spectrum activity against adhered ESKAPE pathogens, namely *E. faecium* (80% CFU reduction), *S. aureus* (95% CFU reduction), *K. pneumoniae* and *E. coli* (complete growth inhibition), *A. baumannii* (40% CFU reduction), and *P. aeruginosa* (98% CFU reduction). This bioactivity was also checked in the presence of serum or high salt concentration against *S. aureus*, resulting in promising activity only in the high salt condition. However, biofilm inhibition of *S. aureus* and *E. coli* during an extended time frame (24 to 72 h) was dependent on the initial inoculum. Nevertheless, FK-16-coated surfaces revealed no toxicity to human epidermal keratinocytes (HaCaT cells). Together, these results demonstrate that shortened versions of LL-37 peptide may be sufficient for antimicrobial activity [99].

Using piranha solution-treated Ti, De Zoya et al. also explored the strategy of silanization with APTES, combined with PEG spacer for immobilization of the AMP GZ3.163 (2 mg/mL). The selected bifunctional PEG spacer (NHS-PEG-MAL) possessed an *N*-hydroxysuccinimide (NHS) group for PEG anchoring to the carboxy-silanized surface, and a maleimide (MAL) group to establish, through the other end, a stable thioether link to the GZ3.163 peptide upon reaction with the cysteine side chain sulfhydryl. The coated samples, with an AMP surface density of 140 ± 18 ng/cm^2^, caused a 98–99% reduction (CFU counting) in bacterial attachment when tested against *P. aeruginosa* and *E. coli* [97]. Chen et al. followed a similar thioether linkage strategy, but using the 4-(*N*-maleimidomethyl)cyclohexane-1-carboxylic acid 3-sulfo-*N*-hydroxysuccinimide ester (Sulfo-SMCC) spacer and a different AMP (melamine, 2 mg/mL) [101]. The coated surface, with an XPS-estimated amount of 3.1 × 10^−9^ mol/cm^2^ of the AMP, significantly decreased both the in vitro bacterial adhesion and biofilm formation by *P. aeruginosa* (62%) and *S. aureus* (84%), in comparison with non-coated Ti surfaces [101]. Importantly, the surface was able to reduce the bacterial load by up to 2 log units in both mouse and rat subcutaneous infection models, when compared to the uncoated surface [101]. These results are promising, since they imply that melimine remained active in the coating after either heat or ethylene oxide sterilization, which can constitute an advantage for its future implementation [101].

Hoyos-Nogues et al. explored a different approach where both antimicrobial and osteointegrative activities were pursued. To this end, a branched peptide platform (named PLATF) was assembled by solid-phase peptide synthesis for parallel display of both the well-known RGD cell adhesive sequence and the lactoferrin-derived AMP hLF1-11 (100 µM). PLAFT was then conjugated, via the *N*-succinimidyl-3-maleimidopropionate (SMP) crosslinker, onto functionalized Ti produced by silanization of nitric acid-treated Ti with APTES [102]. This dual-function surface inhibited the early adhesion of both *Streptococcus sanguinis* (83%) and *S. aureus* (91%) on Ti, while promoting cell integration [102]. These results were further supported in biofilm assays, where PLATF samples reduced the onset of bacterial growth (37% for *S. sanguinis*; 31% for *S. aureus*), as compared to surfaces with direct single hLF1-11 coupling. This is a promising result, considering that PLATF samples presented lower molar density of peptide than that achieved by direct hLF1-11 coupling (PLATF: 13 pmol/cm^2^ versus hLF1-11: 37 pmol/cm^2^), which agrees with the importance of the accessibility of the antimicrobial motifs to bacterial cells [102] already highlighted in other immobilization approaches [97,98,99,101].

Ti silanization strategies were also explored for dental applications. Using another silanization agent, γ-aminopropyltriethoxysilane (APS), Makihira et al. pursued a multistep chemical approach to immobilize JH8194 (20 µM), a derivative from the histatin family [109], onto Ti disks (see Table 1). This surface was capable of inhibiting 80% of *P. gingivalis* biofilm formation, while inducing osteoblast differentiation [103]. Additionally, this strategy was applied on the surfaces of dental fixture in an in vivo canine model (canine mandibles), resulting in the enhancement of trabecular bone formation and acceleration of osteointegration [110].

Holmberg et al. explored the importance of covalently immobilizing the AMP GL13K (0.1 mM) compared to its simple physisorption onto Ti. Starting from alkali-treated Ti, covalent immobilization was pursued using 3-(chloropropyl)-triethoxysilane (CPTES) and diisopropylethylamine (DIPEA) [95]. The GL13K-modified Ti disks reduced in 60% the biofilm formation capacity of *P. gingivalis* and significantly increased mortality of surface-associated bacterial cells in comparison with uncoated Ti disks. Furthermore, an increase in the number of human gingival fibroblasts and MC3T3-E1 osteoblasts within 1 to 5 days post-in vitro incubation on the surface was reported [95].

Later, Zhou et al. applied the same silanization protocol to immobilize GL13K (0.2 mM) onto microgroove Ti, to provide topographic cues to facilitate osteointegration. These surfaces demonstrated excellent bactericidal activity against *P. gingivalis* (75% of the adhered bacterial cells were killed within 24 h), while maintaining biocompatibility towards human gingival fibroblasts and achieving correctly-oriented cell proliferation [96]. Chen et al. applied the same AMP (GL13K, 1 mg/mL) and Ti immobilization strategy, and reported the prevention of biofilm formation by another relevant dental pathogen, *S. gordonii*, under dynamic salivary-flow rate conditions. These authors showed for the first time that GL13K coatings induced bacterial cell wall rupture, thus preventing biofilm formation and growth [104].

### 3.2. AMPs Supramolecular Assemblies

During the last decade, a different strategy has been explored taking advantage of specific material binding sequences, mainly for Ti, which promotes surface peptide immobilization without previous complex surface functionalization [111,112,113]. This strategy stands on use of chimeric peptide constructs comprising both an antimicrobial motif and a Ti-binding domain allowing the exposure of the antimicrobial domain, which remains free to contact and kill bacteria [111]. In this approach, the structural flexibility of the spacer should be considered, as it may determine peptide activity at the interdomain level [113], as well as peptide conformation and hydrophobicity. These properties are interrelated and altogether dictate the efficacy of the AMP domain within a chimeric construct [92,114].

Tamerler and co-workers explored the creation of different chimeric peptides combining three different Ti-binding peptides (TiBPs), TiBPs1–TiBPs3, with an antimicrobial *C*-terminal sequence, either AMP1 (LKLLKKLLKLLKKL) or AMP2 (KWKRWWWWR) [111]. In the interdomain region, a fairly flexible triple glycine (GGG) linker was introduced to enable surface display and, thus, preserve the functionalities of both the TiBPs and the AMPs. The different chimeric peptides were tested, both in solution and after immobilization onto Ti, against *Streptococcus mutans*, *S. epidermidis*, and *E. coli*. Reported results showed no decrease of antimicrobial activity of the immobilized peptides in comparison to the free non-chimeric AMPs. Ti-modified surfaces presented different efficacy depending on the chimeric peptide combination and selected bacteria [111,112]. TiBPs1-AMP1 was the most effective against all tested bacteria, whereas TiBPs2-AMP1 was active against *E. coli* and TiBPs3-AMP2 against *S. epidermidis*.

Similarly, Liu et al. applied the chimeric peptide strategy to the AMP JH8194, which was coupled to a Ti-binding motif (minTBP-1) directly or via flexible/rigid linkers (GGGGS and PAPAP, respectively) for subsequent tethering onto Ti surfaces [113]. The spacer impacted on the final antimicrobial activity of the peptide-grafted surface, with best results achieved when using the rigid linker (minTBP-1 + PAPAP + JH8194), which likely created a clearer separation between domains. This peptide-modified Ti surface presented high bactericidal activity (~90%) against two relevant dental pathogens (*S. sanguinis* and *S. gordonii*), while promoting MC3T3-E1 osteoblasts proliferation and the viability needed for cytocompatibility [109]. Aiming to produce an antimicrobial coating for teeth, Huang et al. [115] developed a hydroxyapatite-binding AMP (HApBAMP), by conjugating the antimicrobial tetrapeptide KSLW to a HAp-binding heptapeptide. The conjugate showed a higher coverage rate on HAp compared to KSLW alone, and a broad range of antimicrobial activity against *S. mutans*, *S. sobrinus*, and *Lactibacillus acidophilus.* Regarding the killing kinetics*,* the surface displayed concentration-dependent and time-dependent bacterial and biofilm killing properties on *S. mutans* (500 μg/mL HApBAMP killed *S. mutans* within 20 min and decreased in 65% *S. mutans* biofilms formation). Moreover, the HApBAMP was cytocompatible with normal human gingival epithelial cells and stable in human saliva. Nevertheless, the antimicrobial activity was still lower than that observed for 0.12% chlorohexidine coating [115].

Another example of a supramolecular assembly-based strategy regards the functionalization of silk proteins with either a fibronectin-binding motif (for cell adhesion) or Magainin I (for antimicrobial activity). The obtained conjugates were capable of forming non-covalently linked coatings on top of different materials (e.g., Ti, HAp) resistant to washing procedures. The reported high level of biocompatibility with fibroblasts and endothelial cells was promising, although the moderate antimicrobial activity against *S. aureus* prompts for further optimizations [116]. Similarly, Zaccaria et al. [117] developed a multi-functional ureido-pyrimidinone (UPy) based supramolecular polymer that was further conjugated with AMPs (SYM11KK, L9K6, or LASIO III) (Figure 5). These supramolecular polymeric materials allowed deposition and self-organization in a film.

However, only the LASIO III-based polymer (4 mol %) produced active surfaces against *E. coli*, Methicillin Sensitive *S. aureus* (MSSA), and MRSA (~2.5 log reduction).

Future clinical application of supramolecular self-assembled or chimeric peptides demands for further research, including assessment of the stability of the surface linkage in a variety of biological conditions (including different pH values, salts levels, and enzymes reflecting distinct organs, tissues, and body fluids), as well as in vivo studies.

### 3.3. AMPs Tethering via Polymeric Systems

The application of polymer structures as implant surface modifiers can convey the improvement of both osteointegration and peptide antimicrobial properties.

Dopamine polymerization into a polydopamine layer supports a variety of reactions, especially due to the catechol group exposed on its surface that can bind covalently to various compounds via Schiff-base or Michael-type addition reactions [118,119].

Xu et al. [120] applied a polydopamine layer onto Ti substrate, followed by covalent attachment of the AMP cecropin B (5 μg/mL). Their goal was to evaluate the impact of cecropin B-based surfaces on antimicrobial activity, osteointegrative potential, as well as the effect on inflammatory response. Cecropin B-coated surfaces were capable of impairing *S. aureus* and *P. aeruginosa* adhesion and promoting a 4-fold or 2-fold viability reduction of surface-adhered Gram-positive (*S. aureus* and *Bacillus subtilis*) or Gram-negative (*P. aeruginosa* and *E. coli*) bacteria, respectively [120]. These surfaces also afforded improved osteoblast proliferation and reduced inflammation responses of macrophages, as suggested by the decreased production of the pro-inflammatory cytokines IL6 and TNF-α [120].

Later, Nie et al. followed a similar strategy using a polydopamine layer onto Ti, where the AMP bacitracin was next conjugated through a carbodiimide-mediated addition-elimination (i.e., condensation) reaction. Thus, the surface produced could decrease *S. aureus* and MRSA colonization by 90%, while promoting the osteogenic differentiation of hBMSCs [105]. This coating was later applied onto Ti rods to be challenged in an in vivo prosthesis infection rat model. Implants exhibited antibacterial activity for up to 3 weeks and osteointegration over 12 weeks, suggesting potential for the prophylaxis against Ti implant-associated infections [121].

Polymer brushes are another example of polymer structures that consist of an assembly of chains polymerized by “surface-initiated atom transfer radical polymerization” (SI-ATRP) onto a solid substrate, allowing the production of a controlled and reproducible coating [106,122]. The brushing of materials has been explored over the years for different applications, as reviewed in references [123,124]. Contrary to other strategies (e.g., silanization), polymer brushes obtained from in situ polymerization offer an increased density of functional groups on their surface, putatively allowing high-density conjugation of AMPs, while also providing a flexible linkage [106,107,108].

Gao et al. [106] investigated the influence of different poly(DMA-co-APMA) copolymer brushes on the immobilization of Tet213 onto silanized Ti, and on the surface antimicrobial activity (details in Table 1). Dependent on the graft density of the brushes, the amount of conjugated Tet213 on the surface varied from 2.6 to 4.5 µg/cm^2^. The maximum peptide density reached was 4.5 µg/cm^2^, which is considerably higher than those reported in other studies [108,120,125,126,127,128], including direct conjugation of peptide onto Ti [129]. Both peptide and chain density on the surface influenced positively the antimicrobial activity of the AMP modified-brushes against *P. aeruginosa* [106]. The same immobilization chemistry was applied to tether the AMP Tet20 onto Ti-wires by Gao et al. [107], demonstrating a 5 log CFU decrease of *P. aeruginosa* and *S. aureus* relative to bare Ti-wires. These authors also reported that their surface had potent antimicrobial activity in vivo in a rat infection model, by showing an 85% decrease in CFUs in 10 out of 14 rats.

The pertinence of the polymer-brush strategy over Ti silanization was studied by Godoy-Gallardo et al. [108]. A Ti surface coated with the AMP hLF1-11 through DMA-APMA copolymer brush achieved a higher decrease both in bacterial adhesion and early stages of biofilm formation by the oral cavity pathogens *S. sanguinis* and *Lactobacillus salivarius*. This effect was attributed to the higher peptide density obtained with polymer brushes (1.3 ± 0.1 – 1.6 ± 0.2 μg/cm^2^), compared to direct silanization (0.9 ± 0.2 μg/cm^2^). The different stability of such surfaces after a 2 h sonication protocol revealed distinct coating resistance, demonstrating, in all cases, some degree of AMP leaching and consequent decrease of the antimicrobial activity. Relevantly, no cytotoxic effects were observed against human fibroblasts when using the polymer brush strategy, suggesting the biocompatibility of the latter [108].

Polymer structures other than brushes have been thoroughly explored for AMP tethering, most of them hydrogels. For instance, Cleophas et al. [130] developed an one-pot photopolymerization process that allowed simultaneous hydrogel formation and AMP immobilization to create a coating on polyethylene terephthalate (PET), commonly used in overdenture implants. They also used crosslinked poly(ethylene glycol)diacrylate-based (PEGDA) hydrogels to immobilize Cys-HHC10 derivatives.

Several soluble HHC10 derivatives with different immobilization orientation or bearing D-amino acids were evaluated, and found to have similar bactericidal effects regardless of the modification introduced. An inverso (D-amino acids based) HHC10 derivative stable in pooled human serum was immobilized via thiol-ene addition at a 10 wt% onto the hydrogel and resulted in a 6 log reduction of Gram-positive *S. aureus* and *S. epidermidis* and Gram-negative *E. coli* [130].

Xie et al. immobilized a GH12-derived peptide onto methacrylate (MA) monomers, to develop an antimicrobial dentin-adhesive polymer. GH12-MA (10% wt) was then polymerized and mixed with crosslinkers to give rise to a dental adhesive hydrogel, capable of decreasing *S. mutans* viability in ~80% [131].

Cai et al. [125] synthesized hydrogels composed by 4-arm PEG with hydroxyl ends functionalized either with thiols (PEG10k-4-SH) or propiolic acid esters (PEG10k-4-PP). These allowed the production of hydrogel meshes where SH-AMP (CKRWWKWIRW) was immobilized via nucleophilic thiol-yne addition. The authors demonstrated that AMP-hydrogel coated Ti containing 0.5 mg/mL of SH-AMP was able to inhibit the growth and viability of Gram-positive bacteria (*S. epidermidis* and *S. aureus*) more effectively than of Gram-negative bacteria (*E. coli* and *P. aeruginosa*), with 3 log reduction for *S. epidermidis* versus 2.5 log reduction for *E. coli* [125]. In addition, AMP-hydrogels exhibited negligible cytotoxicity against 3T3 fibroblasts (viability above 85%) [125].

Steven et al. [132] covalently immobilized the peptide E14LKK, derived from magainin, onto PEGylated polyethylene, one of the materials used in total-joint replacement prostheses. Oxidized poly(ethylene) films were grafted with E14LKK by carbodiimide coupling using either NH_2_-PEG-NH_2_ or NH_2_-PEG-COOH (3400 MW) spacers. Only surfaces with controlled terminal immobilization of the AMP, instead of random grafting through lateral amine groups, caused *E. coli* growth inhibition by at least 3 log compared to controls [132].

Natural polymers have also been widely explored for AMP immobilization [126,128,133,134,135]. In this regard, one of the most popular biopolymers has been chitosan, due to its intrinsic antimicrobial properties [136]. Costa et al. [126] explored the immobilization of hLF1-11 onto chitosan thin films. A disulfide bridge was promoted between the natural cysteine of hLF1-11 and modified chitosan (i.e., with either *N*-acetyl cysteine (NAC), or *O*-(2-carboxyethyl)-O’-(2-mercaptoethyl) heptaethylene glycol spacer (Sp)). Despite the low amount of immobilized AMP at the surface, significant viability decrease was observed on adherent MRSA, particularly when a spacer was employed. Later, the same authors [127] studied the impact of other peptide immobilization parameters, such as orientation (*N*-terminal versus *C*-terminal peptide tethering), spacer length, and spacer flexibility, on the properties of chitosan films grafted with the AMP Dhvar5. For this peptide, a head-to-tail amphipathic AMP, orientation upon immobilization was found of chief importance to achieve higher antimicrobial activity against MRSA. In addition, longer spacers also promoted better antimicrobial outcomes, independently of their flexibility [127].

Sahariah et al. [134] immobilized anoplin to chitosan through copper-catalyzed azide-alkyne cycloaddition (CuAAC) reaction. To that end, chitosan had its OH protected prior to its *N*-azidation, followed by CuAAC reaction with alkyne-anoplin derivatives (at either the *N* or the *C*-terminus). Conjugates with different AMP density were obtained ranging from 12 to 40 peptides per chain. The AMP-chitosan conjugates showed increased antibacterial activity compared to the parent peptide, in particular against Gram-negative bacteria. Thus, for *E. coli*, the Minimum Inhibitory Concentration (MIC) of anoplin–chitosan ranged from 4 to 32 µg/mL, while the MIC value for soluble anoplin was 64 µg/mL [134]. The antimicrobial activity of the anoplin-chitosan conjugates was not directly correlated to the amount of immobilized AMP, presenting strain-dependent results. In addition, anoplin–chitosan conjugates were non-hemolytic [134].

Later, Barbosa et al. [135] further explored CuAAC chemistry for Dhvar5 grafting onto chitosan, following chitosan azidation without extra protective steps, and achieving 50 μmol of AMP per gram of polymer. This Dhvar5-modified chitosan powder was then tested as a coating, revealing higher antimicrobial activity against Gram-positive bacteria (*S. epidermidis* and *S. aureus*, with 60% and 50% more dead adhered bacteria, respectively) and reduced adhesion of Gram-negative bacteria (*E. coli* (30%) and *P. aeruginosa* (70%)), when the AMP was *C*-terminally immobilized. This orientation-dependent antimicrobial activity was further confirmed on anti-biofilm assays (~1.5-fold reduction in biofilm mass in *C*-terminally immobilized Dhvar5). Moreover, this coating showed no cytotoxic effect against HFF-1 cells [128].

Despite the multiplicity of studies applying AMP-containing polymers as surface coatings, this strategy remains to be validated in most cases, since very few studies report in vivo evidence, thus prompting for further research to clarify this research line.

## 4. Conclusions

Important breakthroughs have been obtained over the past two decades regarding AMP-orthopedic based-therapeutics. AMPs long-term stability, mechanical properties, low cytotoxicity, and similar efficacy to antibiotics are among the most relevant issues that still need to be taken in consideration during peptide design. Regarding AMP application, release strategies have evolved to guarantee higher loadings and prolonged releasing periods, without affecting osteointegration [32,51,52,59,61,64,77]. As an example, “smart releasing coatings” can timely target release when needed [61,137,138]. Multiple immobilization strategies have been explored to circumvent AMP bioavailability issues, including avant-garde self-assembling strategies or use of polymer brushes, which apparently allow for higher peptide surface densities [120,123]. The combination of both releasing and immobilization strategies, as already advanced by Townsend et al. [139], appears to be a promising way to succeed in infection prevention in the long term. Combining both strategies, multi-functional antimicrobial surfaces can be created, ensuring the “sterilization” of the surgical site and prevention of implant colonization for extended periods of time.

Clinical translation of these strategies is dependent of a complete preclinical validation, including in vitro validation (simulating more realistic scenarios and timeframe) and in vivo validation (comparing directly to standard care or to a new developed technology).

## 5. Future Perspectives

Further research should demonstrate AMP potential of killing without promoting resistance patterns within bacteria. In addition, equivalence or non-inferiority of AMPs in comparison with antibiotics would fast-track research endeavors towards clinical translation. This high AMP-related efficacy may also be supported by immune-modulation, an AMP feature still poorly explored. As mentioned above, for market-oriented research, a complete pre-clinical validation is mandatory to comply with regulatory entities for future market implementation. Standardized assays should be followed as recommended by the FDA (https://www.accessdata.fda.gov/scripts/cdrh/cfdocs/cfStandards/search.cfm, accessed on 1 September 2021) and EMA (https://www.ema.europa.eu/en/human-regulatory/overview/data-medicines-iso-idmp-standards-overview, accessed on 1 September 2021), including in vivo validation with appropriate animal models, ranging from mice to dogs or goats [73,74,140,141,142,143]. Small animal models, e.g., mice, rats, and rabbits, are cheaper and can provide important information in initial studies. Bigger models, such as dogs, present very similar bone composition and density to humans, and goats or sheep can be good models to mimic human osteomyelitis since their size allows the use of human implants rather than adapted ones [74].

In the coming years, important achievements are expected regarding the development of novel AMP-based strategies, and it is likely that some of them reach the orthopedic devices market/clinics.

## Figures and Tables

**Figure 1 pharmaceutics-13-01918-f001:**
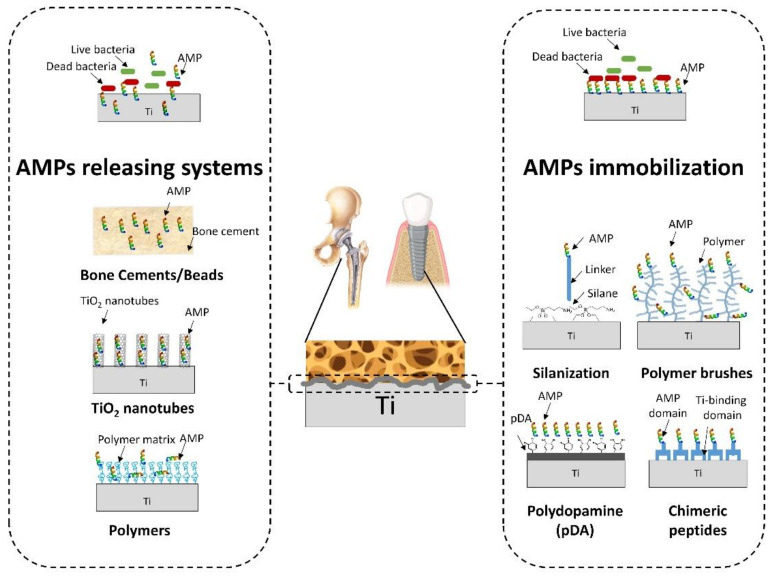
Overview of the currently developed AMPs-based strategies for orthopedic and dental implants.

**Figure 2 pharmaceutics-13-01918-f002:**
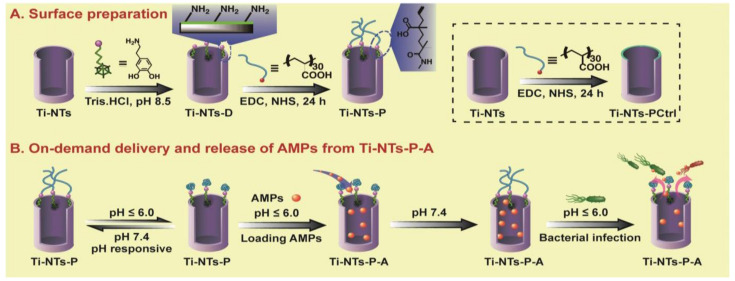
AMPs smart delivery by TiO_2_ nanotube system. (**A**) Surface preparation. (**B**) On-demand AMPs’ delivery from the system. Titanium nanotubes (Ti-NTs), titanium nanotubes modified with dopamine (Ti-NTs-D), PMAA “gates” engineered Ti-NTs (Ti-NTs-P), PMAA “gates” engineered Ti-NTs with encapsulation of the loaded AMPs (Ti-NTs-P-A), and control groups (Ti-NTs-PCtrl). Adapted with permission from reference [61].

**Figure 3 pharmaceutics-13-01918-f003:**
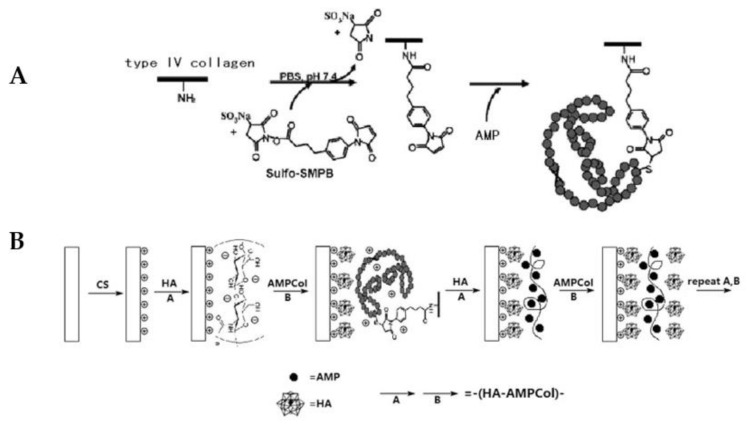
Schematic representation of Chitosan and Hyaluronic Acid Layer by layer coating production embedding an AMP-collagen IV conjugate. (**A**) Synthesis of AMPCol; (**B**) CS-(HA-AMPCol) layer by layer coating production. Adapted with permission from reference [64].

**Figure 4 pharmaceutics-13-01918-f004:**
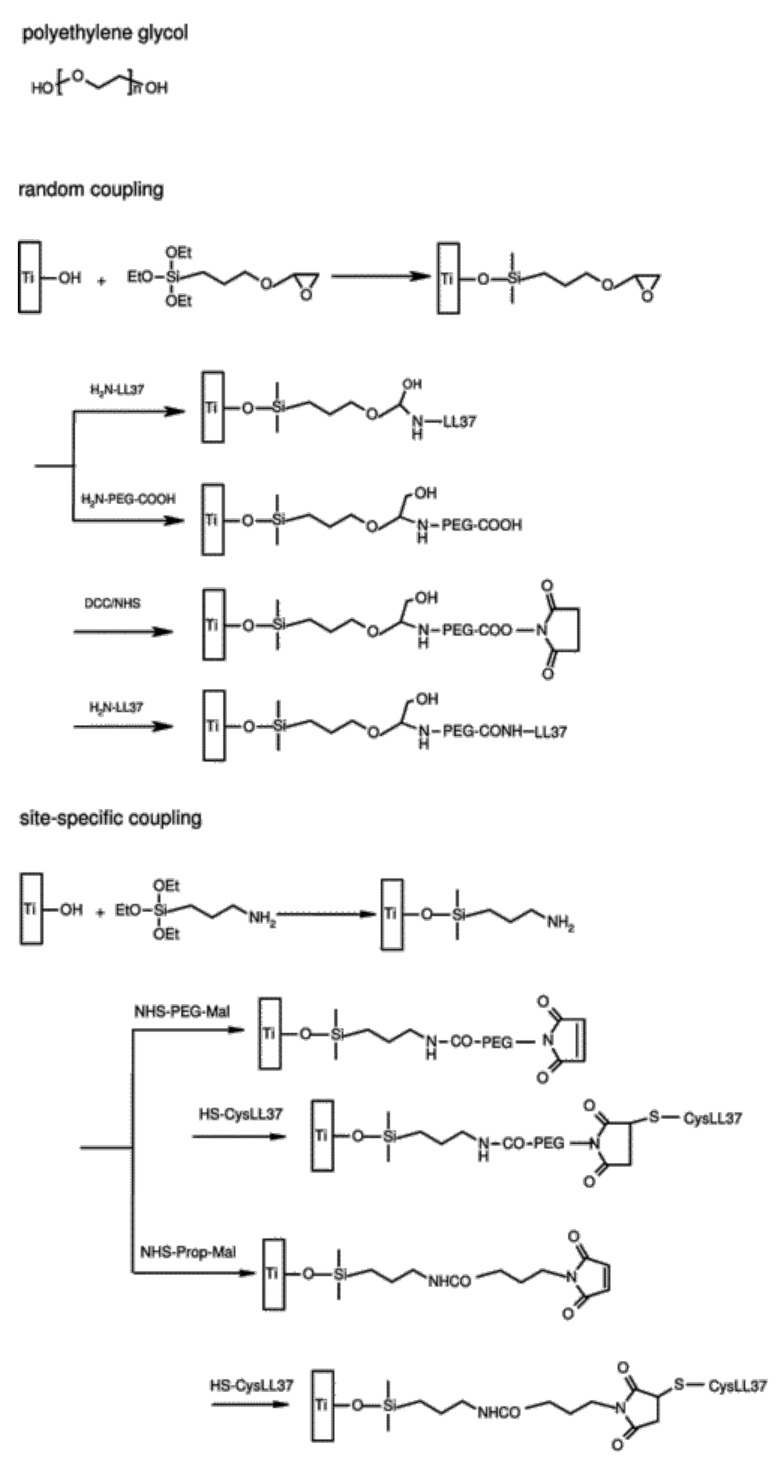
Reaction schemes of AMP conjugation following random and site specific coupling methods with or without a spacer introduction onto silanized titanium. Adapted with permission from reference [98].

**Figure 5 pharmaceutics-13-01918-f005:**
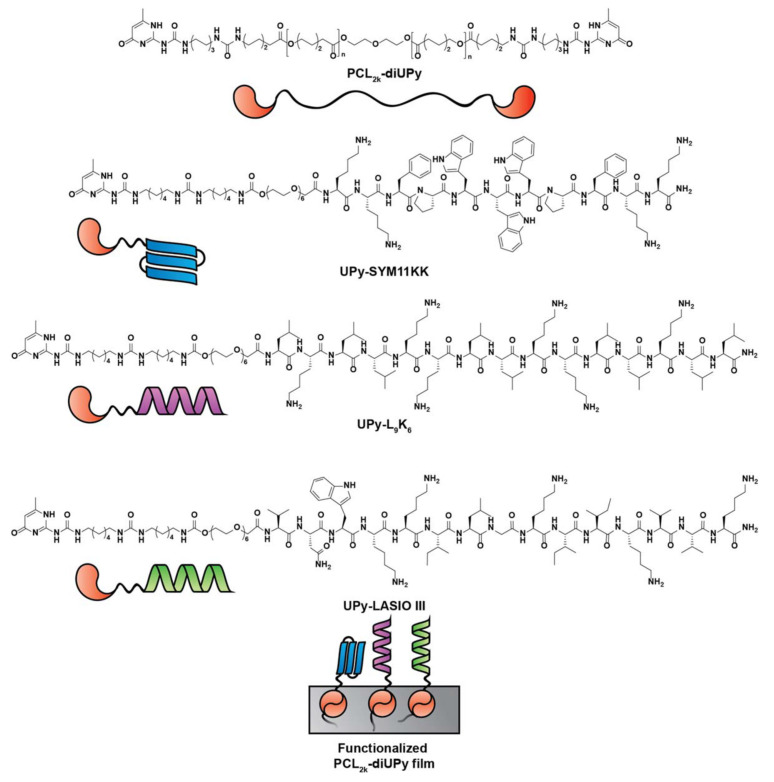
Schematic representations and molecular structures of the UPy-polymer and UPy-functionalized antimicrobial peptides SYM11KK, L9K6, and LASIO III to be applied as a supramolecular assembly-based strategy. Adapted with permission from reference [117].

**Table 1 pharmaceutics-13-01918-t001:** Overview of AMPs’ immobilization-strategies onto Ti surfaces through either silanization or polymer brushes, and their biological performance.

Substrate	AMP	Immobilization Strategy	Microorganisms Assessed	In Vitro Testing	In Vivo Testing	Biocompatibility	Ref
Ti squares	LL-37	1. Piranha solution 1 h 2. Silanization with APTES^1^ (2%, in toluene, 18 h, RT) 3. PEG linker: NHS-PEG-Mal 4. Incubation with peptide	*E. coli* (strain K12)	Bacterial killing assay (Propidium iodide staining)	n/a	n/a	[98]
Ti squares	FK-16 (fragment 17-32 of LL-37)	1. Etching with 5 M NaOH (24 h, 60 °C) 2 Silanization with APTES^1^ (0.5% in anhydrous toluene, 1 h, 70 °C) 3. Crosslinker 6-maleimidoheaxanoic acid 4. Incubation with peptide	*E. faecium* (ATCC51559) *S. aureus* (USA300) *K. pneumoniae* (ATCC13883) *A. baumannii* (B2367-12) *P. aeruginosa* (PAO1) *Enterobacter cloacae* (B2366-12) *E. coli* (ATCC 25922)	XTT assay, CFU assay	n/a	HaCaT cells Hemolysis assay	[99]
Ti foil	GZ3.163	1. Piranha solution (30 min) 2. Silanization with APTES^1^ (1% in dry toluene, 16 h) 3. PEGylation with NHS-PEG_24_-MAL 3 ester 4. Incubation with peptide	*P. aeruginosa * (ATCC 27853) *E. coli* (DH5α)	CFU assay, BacLight viability assay, SEM	n/a	Mouse blood cells lysis assay	[97]
Ti disks	Melimine	1. Piranha solution (2 min) 2. Silanization using APTES^1^ (10% *w*/*v* in dry toluene, 1 h, RT) 3. Crosslinker sulfo-SMCC^2^ 4. Incubation with peptide	*S. aureus * (strain 38) *P. aeruginosa * (PAO1)	BacLight viability assay	Mouse and rat sub-cutaneous infection models, CFU assays	n/a	[101]
Ti disks	LF1-11	1. Etching with nitric acid 65% *v*/*v* (1 h) 2. Silanization using APTES^1^ (2% *v*/*v* in anhydrous toluene, 1 h, 70 °C) 3. Crosslinker *N*-succinimidyl-3-maleimido-propionate 4. Incubation with peptide	*S. aureus* (CCUG 15915, Sweden) *S. sanguinis* (CECT 480, Spain)	BacLight viability assay, SEM	n/a	SaOS-2	[102]
Ti disks	JH8194	1. Etching with 10 mM NaOH (24 h) 2. Silanization with APS^3^ (5% in acetone, 15 min) 3. EDC/NHS^4^ 4. Incubation with peptide	*P. gingivalis * (oral cavity isolate)	Growth curves	n/a	MC3T3-E1	[103]
Ti disks	GL13K	1. Etching with 5 M NaOH (ON, 60 °C) or treated with O_2_ plasma (5 min) 2. Silanization with CPTES^5^ and DIPEA^6^ (RT, 1 h) 3. Incubation with peptide	*P. gingivalis* (ATCC 33277)	ATP assay, CFU assay	n/a	HGF, MC3T3-E1	[95]
Micro-structured silicon substrate plus a layer of Ti	GL13K	1. Etching with 5 M NaOH (30 min, 60 °C) 2. Silanization with CPTES^5^ and DIPEA^6^ (1 h, RT) 3. Incubation with peptide	*P. gingivalis* (ATCC 33277)	CFU assay, Acridine orange staining Assay	n/a	HGF	[96]
Ti disks	GL13K	1. Etching with 5 M NaOH (60 °C, ON) 2. Silanization with CPTES^5^ and DIPEA^6^ (1 h, RT) 3. Incubation with peptide	*S. gordonii* (strain ML-5)	Drip Flow Bioreactor Culture, CFU assay, ATP Assay, BacLight viability assay, SEM	n/a	n/a	[104]
Ti substrates	KR-12 (fragment 18–29 of LL-37)	1. Etching with 5 M NaOH (24 h, 80 °C) 2. Silanization with APTES^1^ (5% in hydrous toluene, 8 h, RT) 3. Incubation with peptide	*S. aureus* (ATCC 25923) Methicillin-resistant *S. aureus* (MRSA, ATCC, 43300) *S. epidermidis* (ATCC 35984) Methicillin resistant *S. epidermidis* (MRSE, ATCC) *E. coli* (ATCC 25922)	CFU assay, BacLight viability assay, SEM, CLSM	n/a	hBMSCs	[105]
Ti deposited silicon wafers	Tet213	1. Silanization with APTES^1^ modified using glycidol 2. Surface-Initiated ATRP of DMA^x^ and APMA^y^ 3. Maleimide group grafting 4. Incubation with peptide	*P. aeruginosa* (PA01 expressing a luciferase gene cassette (luxCDABE))	CFU assay, lumi-nescence	n/a	n/a	[106]
Ti deposited silicon wafers	Tet20	1. Silanization with APTES^1^ modified using glycidol 2. Surface-Initiated ATRP of DMA^x^ and APMA^y^ 3. Maleimide group grafting 4. Incubation with peptide	*P. aeruginosa* ( ATCC 27853) *S. aureus* *P. aeruginosa* (PA01 expressing a luciferase gene cassette (luxCDABE))	CFU assay, lumi-nescence, SEM	Rat sub-cutaneous infection model	MG-63, Platelet activation, Complement activation analysis	[107]
Ti cylinders	hLF1-11	1. Silanization with either APTES^1^ or CPTES^2^ 2. Surface-Initiated ATRP of DMA^x^ and APMA^y^ 3. Et_3_N 4. Incubation with peptide	*S. sanguinis* (CECT 480) *L. salivarius* (CCUG 17826)	CFU assay, BacLight viability assay , CLSM, BacTiter-Glo biofilm assay	n/a	HFF	[108]

^1^ APTES: 3-Aminopropyl triethoxysilane; ^2^ Sulfo-SMCC: 4-(*N*-maleimidomethyl)cyclohexane-1-carboxylic 3-sulfo-*N*-hydroxysuccinimide ester; ^3^ APS: γ-aminopropyltriethoxysilane; ^4^ 1-Ethyl-3-(3-dimethylaminopropyl) carbodiimide (EDC)/*N*-hydroxysuccinimide (NHS); ^5^ CPTES: (3-chloropropyl)triethoxysilane; ^6^ DIPAE (diisopropylethylamine; n/a: not applicable, Et_3_N: Triethylamine, CFU: Colony Forming Units, SEM: Scanning Electron Microscopy, CLSM: Confocal laser scanning microscopy. ^x^ *N*,*N*-Dimethylacrylamide; ^y^
*N*-(3-Aminopropyl)methacrylamide Hydrochloride; RT: room temperature; ON: Overnight.

## Data Availability

Not applicable.

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
