# Peer review of "Antimicrobial Peptides in the Battle against Orthopedic Implant-Related Infections: A Review"

_pharmaceutics, 2021, doi:10.3390/pharmaceutics13111918_

Round 1

Reviewer 1 Report

This review paper summarizes the research related to the use of antimicrobial peptides for application in orthopedic implant related infections, which is an interesting and important area of research. The paper is generally well written and merits publication; however, the quality of the paper can be enhanced if the following points can be addressed.

  1. Does the metals used in orthopedic implants play any role in the efficacy of antimicrobial peptide antimicrobial activity? If yes, please briefly mention about this.
  2. The authors have mentioned limited drug reservoir capacity and bioavailability as a limitation for AMP release systems, however it would be better to discuss what stratergies are employed or under research to overcome this shortcoming?
  3. Do bone cements have any disadvantages or limitations ?
  4. Will the metal composition have any influence on the bioaailability or antimicrobial activity of AMP's
  5. Authors mentioned about silanol functionalization. There were seeral reports on the decline of biomolecule efficacy up on interaction with silanes. Are there any reports that compared the efficacy of free AMPs and silane linked AMP's
  6. .There is always a dilemma on how to conclude a review article. Since the authors have deliberately summarized huge amounts of published results, it will go a long way. It would be helpful if they can provide their own thoughts that would in turn help in finding the areas that need to be addressed. For example, what are the factors that one needs to consider while desiging AMP implants, what are the required criteria to overcome the toxicity/limited bioavailability of AMP s and what are the steps required for the fast transition of these materials for industrial scale up. In general, what measures need to be taken for the effective clinical translation of AMP based implants?  What limitations are hindering their clinical translation and in what direction does the future research need to be, to make the clinical translation possible?

Reviewer 2 Report

The Review entitled “            Antimicrobial peptides in the battle against orthopedic implant-related infections” is original and has significance for the scientific community. Authors have analyzed 135 literature sources including their own research publications that indicate their awareness of this issue. The presented review focuses on the antimicrobial peptides (AMP) that may prevent the orthopedic and dental implant-related infections. Authors highlighted the major strategies such as AMP-releasing systems from titanium-modified surfaces along with AMP immobilization.

Introduction section reveals the relevance of the selected problem describing antimicrobial peptides as agents that may overcome antibiotic resistance. The major strategies for orthopedic and dental implant-related infections are displayed at the second and third section. As a reviewer, I have no additional remarks or correction to the Manuscript since it is properly structured and fully reveals the essence of the problem.  

Therefore, I highly recommend to accept this manuscript for publication at the present form.

Reviewer 3 Report

Antimicrobial peptides in the battle against orthopedic im-2 plant-related infections – a review

 The current review focuses on the potential of using AMPs as a preventive strategy for 24 orthopedic and dental implant-related infections.  No doubt the subject addressed is important and has gained significant attention in the research area, however, the extent of the effectiveness of this work need to be elucidated. Many terms are abbreviated without giving complete form. It is advised to use the complete name of that term when used firstly. This manuscript needs revisions before could be considered for publication. Therefore, the authors are advised to address the following comments carefully.

  • The abstract does not fully encompass what has been described in the text. Please further develop the abstract.
  • Self-contained abstract of approximately 200 words, outlining in a single paragraph the aims, scope, and conclusions of the paper.
  • The introduction needs to be more emphasized on the research work with the detailed explanation of the whole process considering past, present, and future scope.
  • The conventional methods to be explained well to indicate the relevance of the research work. It needs to be strengthened in terms of recent research and updated literature review in this area with possible research gaps.
  • It is strongly recommended to add a recent literature survey about antimicrobial peptides in the battle against orthopedic im-2 plant-related infections.
  • Research gaps should be highlighted more clearly, and future applications of this study should be added.
  • My main concerns are about the lack of backup citations for some key claims and statements throughout the review article.
  • Conclusions given here need to be rewrite. It may be remembered that this Section forms a summary of all the major observations/ results obtained. Accordingly, here presentation should consist of the main Results or the observations of the study in a short paragraph. Hence better to rewrite/rephrase this Section.
  • In the Future perspectives section, it should be clearly presented what should be done or what kind of future research should be carried out to allow real-time practical application.
  • Reviews should normally contain a majority of new artwork. As a guideline at least 70% of the artwork in a review should be newly created. Please draw more Figures to illustrate your work.

Round 2

Reviewer 3 Report

All requested adjustments were accepted by the authors of the manuscript. I recommend accepting the paper.